# Ship Berthing and Unberthing Monitoring System in the Ferry Terminal

**DOI:** 10.3390/s22239133

**Published:** 2022-11-24

**Authors:** Teresa Abramowicz-Gerigk, Jacek Jachowski

**Affiliations:** Faculty of Navigation, Gdynia Maritime University, 81-225 Gdynia, Poland

**Keywords:** monitoring system, ship berthing, seabed protection, bow thruster jet, CFD simulation

## Abstract

This paper presents a monitoring system designed to increase the safety of the quay structure in ferry terminals, in which, during berthing and unberthing maneuvers, propeller and thruster-generated jets may damage the seabed protection, threatening the stability of the berth structure. Direct measurement of flow velocity on the seabed is not possible due to the possibility of its damage, therefore dynamic pressure measurement of the quay wall was used within the system. The relationship between the pressure on the quay wall and flow velocity on the seabed was determined using real-scale CFD simulation of the flow field generated during berthing and unberthing maneuvers. The paper focuses on the computations of the pressure distribution generated by bow thrusters. These computations made it possible to determine the velocity field in the time domain. Their results, verified using real-scale measurements, are in line with generally accepted empirical methods.

## 1. Introduction

The impact of bow thruster jets on seabed protection over the long term may cause damage, which may result in the loss of stability of the quay structure. The monitoring system presented in the paper allows for the detection of exceedance of allowable loads.

The system should be designed for particular berths and ships. In the case of ferries, the most important design parameter is the position of bow thrusters in the last phase of berthing or at the beginning of an unberthing maneuver. In both cases, when the speed of the ship is very low or the ship is stationary, strong jets are induced on the quay wall and the seabed.

In operational practice, speed limits are translated into recommendations regarding the used power of devices. The recommendations are followed by ship masters; however, they can be exceeded in weather conditions close to the ferry operational window during operations carried out without tugboat assistance.

The pressure sensors of the monitoring system could not be installed on the seabed due to their possible damage, therefore the determination of bed velocities was based on the measurement of dynamic pressure on the quay wall.

The presented system was installed in the ferry terminal in the Port of Gdynia. The relationship between the maximum flow velocities on the seabed and pressure on the quay wall was determined using empirical methods and simplified numerical computations.

The studies on the prediction of the flow generated by a bow thruster near the quay wall, based on CFD simulation using simplified ship models and verified by experiments in small scale, were studied by Van Blaaderen [1] and Van den Brink [2].

Their main conclusions were related to the influence of the geometry of the bow thruster on the flow field and velocity distribution on the seabed. The correct modelling of the ship geometry in the area of the bow thruster is important in the simulation of the induced flow and was recognized as the most important aspect in numerical modelling [1].

In the presented study, the real ship hull form is modelled, including bow thruster models in the form of virtual propellers inside the cylinders, with inlet and outlet openings diameters equal to the propeller ducts.

The real-scale modelling enables an in-depth recognition of the phenomena and allows the determination of flow distribution in the space–time domain which then can be used in prediction of a possible seabed protection failure.

The real-scale measurements of bow thruster-generated pressure on the quay wall were used for 3D numerical model validation.

One of the problems related to numerical simulation is use of the proper turbulence models. They are different from the physical models because of the assumption of isotropic turbulence. Van Blaaderen [1] and Van den Brink [2] presented the numerical studies of CFD modelling of the thrusters operating at quay walls using the k-ε turbulence model. The sensitivity analysis of the numerical model presented by Van Blaaderen [1] concluded that the k-ε turbulence model is justified and assumption of roughness also has a little influence on the flow field. In the presented study, based on the previous investigations, the LES (large eddy simulation) turbulence model was used [3].

The main objective of the presented research was to determine the bow thruster generated velocity field on the seabed on the basis of pressure measurement on the quay wall in time domain. The generally accepted empirical methods used in design practice of the berth structures, seabed protections and development of operational guidelines were used to verify the numerical simulation results [3,4,5,6].

## 2. Materials and Methods

The list of variables used in the paper is presented in Table 1.

### 2.1. Measurement of Bow Thruster-Generated Pressure on the Quay Wall

The measuring system was designed for the Stena Baltica ferry (Ferry 1) and then used for the 10 m longer Stena Vision ferry and her sister ship, Stena Spirit ferry (Ferry 2), put in operation after the system was installed. Ferry 1 was equipped with the bow and stern thrusters, Ferry 2 had two bow thrusters of the same size.

The main particulars of the ferries are presented in Table 2.

The measurements were collected using the system of pressure meters installed in 8 vertical lines with the separation distances 1.2 m with 8 pressure meters in each line. The measuring system has been adapted to be easily moved to another place, therefore the pressure sensors have been screwed to the pipes, consisting of segments folded for transport [7]. The supporting structure was attached to the quay wall, made of the Larssen sheet piles with screw pins. The 131S (II) BCM Sensor Technologies piezo-resistive pressure transmitters (www.bcmsensor.com, accessed on 27 October 2022) were integrated with the microprocessor control system and communication module [8].

The measuring range of the system was 0–300 kPa with 0.1% FS. The assumed minimum value recognized by the measuring system with the satisfactory level of measurements repeatability was 100 Pa. The accuracy of measurement was 100 Pa with 0.1% relative error. The frequency of measurements was 2 Hz and synchronization deviation of the system was 5 ms.

The layout of sensors of the measuring system installed on the quay wall is presented in Figure 1.

### 2.2. CFD Simulation

The purpose of CFD simulation was to estimate the impact of the jets generated by bow thrusters on the quay wall and seabed. The simulations were carried out using the FLOW-3D program on the workstation with Intel Xeon gold 6244 processor, 3.60 GHz processor and RAM 96 GB.

CFD modelling was carried out using the full-scale (real ship) in 6 DOF (degrees of freedom) in steady state, with a stationary ship, using overlapping rectangular grids with 5.5 mln cells.

The geometry of the maneuvering area, hull form model of Ferry 2, grid of pressure sensors and overlapping meshes applied for the simulations are presented in Figure 2.

The computing domain and boundary conditions applied for the flow field distribution simulations are presented in Figure 3. The dimensions of the computing domain were 260 m × 120 m × 14 m.

The assumed water density and kinematic viscosity were equal to *ρ* = 1005 kg/m^3^ and ν = 10^−6^ m^2^/s. The large eddy simulation (LES) turbulence model and VOF (volume of fluid) method for the free surface modeling were applied. A variable time-step dependent on the convergence of a solution was used.

The thrust values of bow thrusters were estimated on the basis of the given maximum power of bow thruster drives. They were modelled using the FLOW-3D software on the basis of the measured velocity field at the bow thruster outflow opening and calculated average flow rate.

The parameters of the virtual bow thruster modelled using the FLOW-3D software are presented in Table 3.

Figure 4 presents the bow part of the hull form with bow thruster models. The outflow velocities were calculated for the assumed 105 kN thrust for each propeller.

Figure 5 presents the thrust of the modelled bow thruster in dependence on the revolutions of the virtual thruster propeller.

## 3. Results and Discussion

### 3.1. Bow Thruster Generated Pressure on the Quay Wall

Most of the measurements during the ferries normal operation showed the small values of bow thruster generated pressure, no greater than 3 kPa. The highest-pressure values up to 8 kPa were collected for Ferry 1 [8]. The bow thruster openings projections of Ferry 1 in moored position were between the C3, C4 and C6, C7 measuring lines. The bow thrusters’ projections of Ferry 2 were between the C5, C6 lines and just after the C8 line (Figure 1). During the CFD simulation, the positions of bow thrusters’ openings projections were between the C3, C4 and C6, C7 measuring lines. In the case of Ferry 2, the C3, C4 lines on the quay wall correspond to the C5, C6 lines using the CFD simulation and the C6 line on the quay wall corresponds to the C8 line using the CFD simulation.

An example of the time series of pressure measurements for Ferry 1, presented in Figure 6a, can be compared with the values calculated using the CFD simulation shown in Figure 6b. Both the results are presented within the C5 measuring line. Within the simulation model, the assumed 130 kN thrust produced by both bow thrusters during the period of time of 60 s was used.

The comparison of the time series of pressure collected on the quay wall for Ferry 2 with the results of numerical simulation is presented in Figure 7. In this case, the C8 line on the quay wall is corresponding to the C6 line within the simulation.

The results of simulation were plotted with the 20 Hz frequency showing the single pulses of higher pressure. The average maximum pressure calculated was close to the measurements. The differences between the measurement and simulation results are related to the real conditions of measurements when a ship starts to move sideways and forward and different time of maximum power is used. They are also related to the frequencies of measurements in real-scale equal to 2 Hz.

The results of CFD simulation for Ferry 2, with both the bow thrusters operating with 105 kN thrust each during the period of time of 5 s, are in line with the measurements collected during the berthing of Ferry 2 presented in Figure 8a. Figure 8b presents the maximum absolute values of pressure within the C3–C8 lines.

The maximum computed values are in line with the measurements, the maximum single pressure pulses are two times bigger than the measurements.

### 3.2. Velocity Field on the Quay Wall and Seabed

The time series of bow thruster jets generated pressure and flow velocity on the quay wall in the C7 line, computed for both the bow thrusters operating with 105 kN thrust each during the period of time of 10 s, is presented in Figure 9.

The velocities over 2 m/s appeared after the simulation time of 15 s. The maximum velocities 2.4 m/s appeared after the time of 20 s and after the time of 25 s they decreased below 1.5 m/s. The maximum calculated velocity on the seabed at the position of the C7 8 pressure meter was equal to 2.4 m/s.

The related velocity field distribution on the seabed for Ferry 2 is presented in Figure 10. In Figure 10a, both the bow thrusters operating with 105 kN thrust each during the period of time of 10 s generated maximum velocity 2.4 m/s.

In Figure 10b, both the bow thrusters operating with the higher thrust equal to 130 kN each during the period of time of 10 s generated maximum velocity 3.0 m/s.

The bow thruster-generated flow distribution on the quay wall and seabed described by Schmidt [9] was divided into five zones including flow on the quay wall—zone 3 and, on the seabed, zone 5. The scheme of the flow distribution is presented in Figure 11a [8].

The bow thruster jet on the quay wall and seabed, observed within the CFD simulation, is presented in Figure 11b–d for the bow thrusters operating with the thrusts equal to 105 kN, 130 kN and 160 kN during the period of time of 10 s, respectively. The results of CFD simulation illustrate the flow velocity distribution in Zone 5 on the seabed equal to the jet velocity in front of the quay wall.

The decrease of the flow velocity when the flow passes from zone 3 to zone 5, according to Römisch, can be neglected [10]. The relationship between the dynamic pressure on the quay wall and velocity on the seabed (Equation (1)) can be calculated based on Bernoulli equation (Equation (2)), neglecting water viscosity.
(1)$$p=0.5\;\rho v^{2}$$
where v [m/s] is the flow velocity, *ρ* [kg/m^3^] is the water density.
(2)$$p+0.5\;\rho v^{2}+\rho \mathrm{gh}=0$$
where h [m] is the water depth.

The towing tank tests showed 5% absolute relative percentage error between the pressure values measured at various speeds and calculated using Equation (1).

The flow velocity distribution presented in Figure 11 shows the maximum velocities generated on the quay wall and seabed after 20 s of simulation. The flow velocity distribution in time in dependence on the thrust applied to the bow thruster is presented in Figure 12 and Figure 13.

The bow thruster-generated flow distribution in the section of measuring C7 line during the period of time, respectively, 15 s, 20 s and 25 s for both the bow thrusters operating with the thrusts of 105 kN, 130 kN and 160 kN each, is presented in Figure 12.

To determine the flow distribution with the maximum velocity values on the quay wall and seabed, the simulation time should be long enough. In the studied case, this time is about 20 s.

The bow thruster-generated flow distribution in the bow thruster symmetry plane between C3 and C4 lines during the period of time, respectively, 10 s, 15 s, 20 s and 25 s for both the bow thrusters operating with the thrusts of 105 kN, 130 kN and 160 kN each, presented in Figure 13, shows that during the first 10 s of simulation the maximum velocities are generated on the seabed only; then the jets reach the wall.

The range of the flow velocity presented in Figure 12 and Figure 13 was assumed as 3.5 m/s to clearly present the flow distribution on the wall and seabed. The real efflux velocities are higher. Their mean values are equal to 5.3 m/s, 5.7 m/s and 6.7 m/s for the thrust values 105 kN, 130 kN and 160 kN, respectively. The efflux velocities are presented in Figure 14.

The results of CFD simulation show the bow thruster-generated flow velocity on the seabed in time. In the studied cases of Ferry 2, the bow thrusters operating with 105 kN thrust each during the period of time of 5 s generated the pressure no greater than 1.4 kPa which after 5 s dropped to 0.5 kPa. The bow thrusters operating with 105 kN thrust each, during the period of time of 10 s, generated the maximum pressure value of 2.4 kPA.

The axial velocity of the jet induced by the bow thruster can be calculated on the bases of bow thruster power by Equation (3).

The maximum velocities can be calculated according to the German method by Equation (4) and the Dutch method by Equation (5) [4].
(3)v0=1.15·NBTρ·DP213
where v_0_ [m/s] is the axial velocity of the jet induced by the bow thruster, N_BT_ is the bow thruster power, D_P_ is the bow thruster propeller diameter, *ρ* = 1000 kg/m is the water density.
(4)$$v/v_{0}=1.9\left(\frac{D_{P}}{L}\right)$$
where L = 0.4 B [m] is the distance between the quay wall and bow thruster opening.
(5)$$v/v_{0}=2.8\cdot \left(\frac{D_{\mathrm{BT}}}{z+L}\right)$$
where z [m] is the height of bow thruster axis over the seabed.

The results obtained using both the methods for Ferry 1 and Ferry 2, compared with the measurements and results of CFD simulation, are presented in Table 4. The axial velocities used within the empirical methods are calculated from Equation (1) for the bow thrusters first from the bow. The CFD simulation results of v_max_ are related to the axial velocity calculated by Equation (3). They correspond to the results obtained for the bow thrusters operating with 160 kN during the period of time of 10 s, in C3 line for Ferry 1 and C6 (position of C3 line on the quay wall) for Ferry 2.

The axial velocities and mean of maximum values of velocities at the lines situated close to the bow thrusters: C3, C4, C6 and C7 obtained from the CFD simulation, compared to the values calculated by the empirical German and Dutch methods are presented in Table 5.

The presented results are between 2 m/s and 2.6 m/s and show a good agreement between the simulation and empirical prediction of the maximum values.

## 4. Conclusions

Thruster jets monitoring increases the safety of the berth structure and decreases the maintenance costs. It allows for the detection of all the emergency situations when a ship’s master uses the power of thrusters above the permissible limits.

The presented CFD simulation in full-scale, based on the real hull form of the ship, and both the bow thruster models including the bow thruster propeller parameters, is a reliable tool in prediction of the bow thruster loads on the quay wall and seabed protection. It has been validated using the data from the monitoring system installed in the ferry terminal. The results were also verified by generally accepted empirical methods. They are close to the measurements; however, they are still estimations due to the different conditions of real maneuvers and possibility of modelling a similar vessel.

The main advantage of the presented method is the possibility of analysis of the bow thruster jet-generated flow field and prediction of possible damage of the seabed protection in the space–time domain.

The accuracy of CFD simulations can be accepted in design of new monitoring systems—prediction of pressure limits, position and layout of the grid of pressure sensors on the quay wall in dependence on the seabed protection characteristics.

The problem of thruster jets impact on the port facilities presented for the ferry terminals is also important for the large cruise vessels equipped with the high-powered thrusters and large containerships which use their thrusters to support the tugboats during the berthing and unberthing operations [11,12,13,14].

The presented approach can be also used in modelling the thruster and propeller jets’ impact on the offshore structures.

The application of the presented monitoring of the condition of seabed protection is especially important for the protection in the form of a mattress of geotextile sand-filled bags. It can save the maintenance costs related to the replacement of a wide area of damaged elements and allows avoidance of the maintenance delays in operation of the berth.

## Figures and Tables

**Figure 1 sensors-22-09133-f001:**
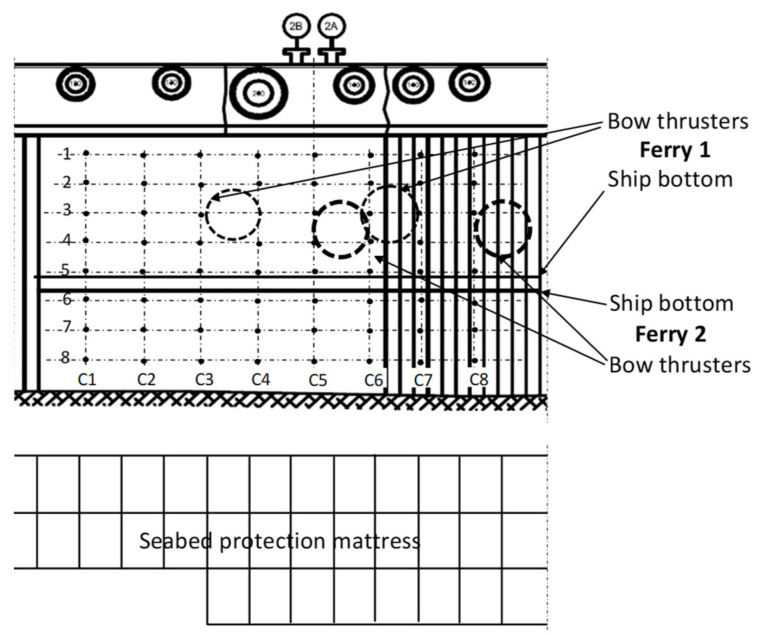
Layout of measuring sensors and projection of the bow thruster openings of Ferry 1 and Ferry 2 moored at the berth.

**Figure 2 sensors-22-09133-f002:**
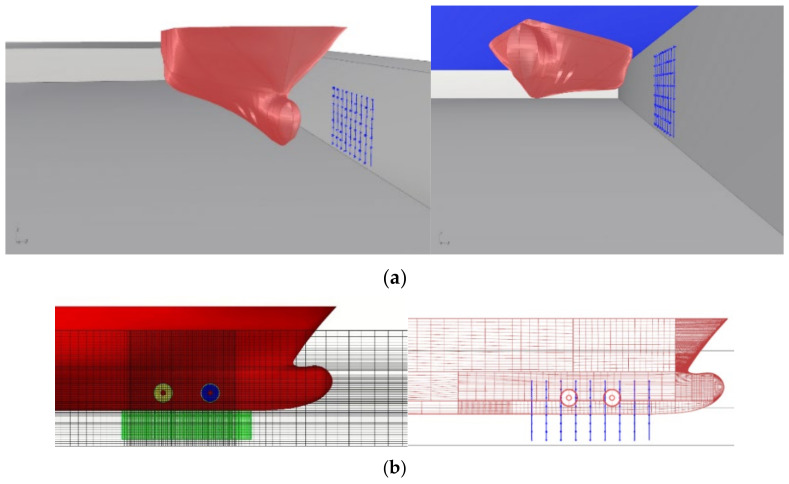
Model of the quay wall with the pressure sensors positions: (**a**) geometry of the maneuvering area, hull form model of Ferry 2 and grid of pressure sensors positions; (**b**) overlapping meshes applied for the simulations.

**Figure 3 sensors-22-09133-f003:**
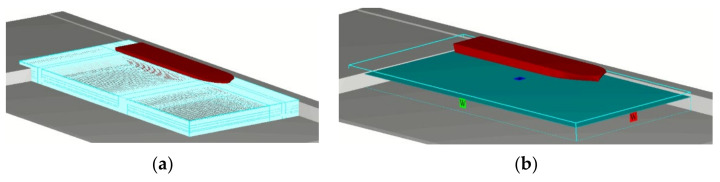
Computing domain: (**a**) computing domain with the overlapping meshes; (**b**) boundary conditions applied for the bow thruster-generated flow field simulation.

**Figure 4 sensors-22-09133-f004:**
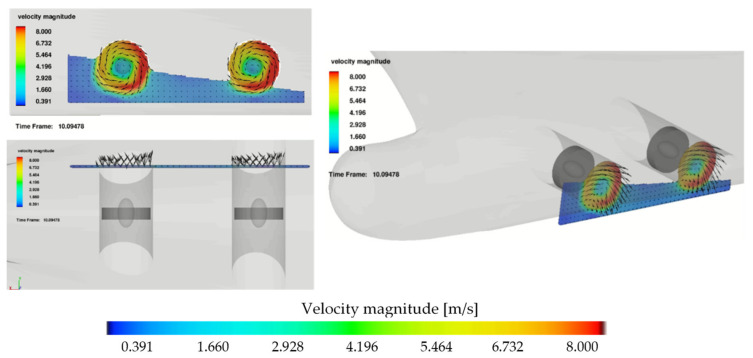
Numerical models of the bow thrusters.

**Figure 5 sensors-22-09133-f005:**
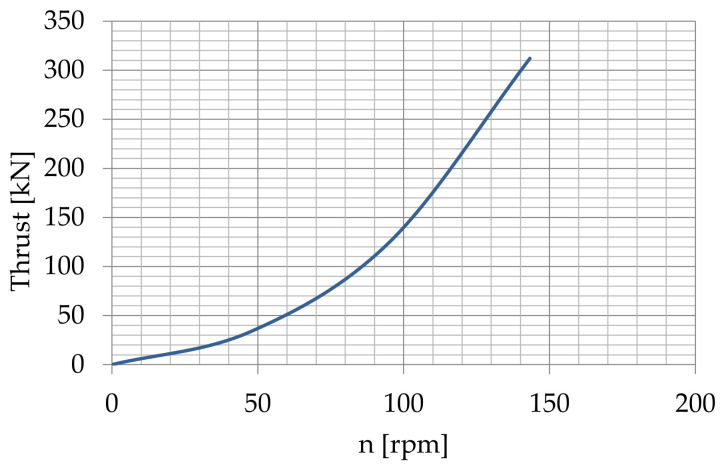
Thrust of the virtual bow thruster in dependence on the bow thruster propeller revolutions.

**Figure 6 sensors-22-09133-f006:**
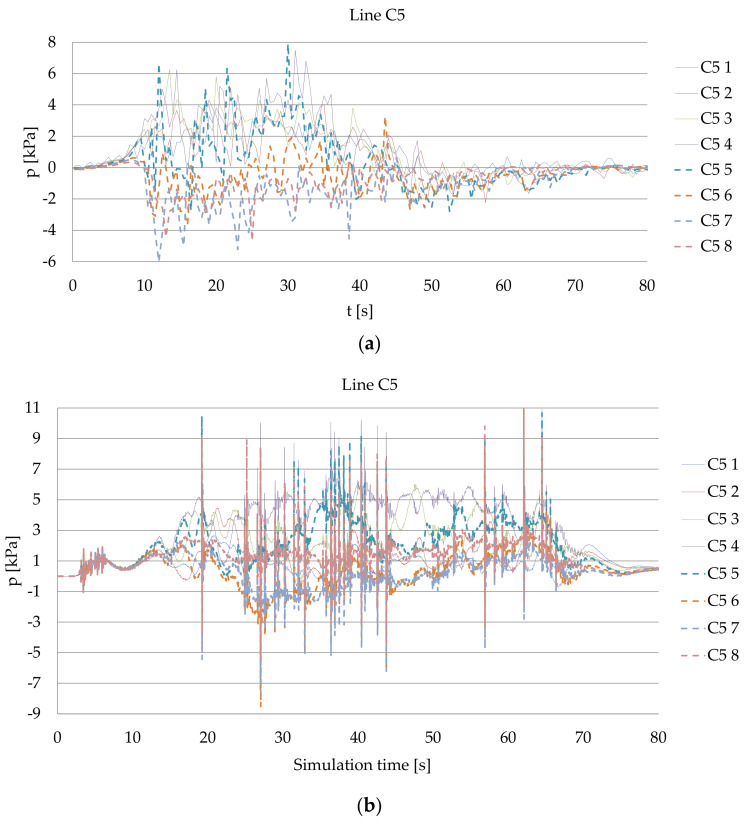
Pressure within the C5 line during the unberthing: (**a**) real measurements for Ferry 1; (**b**) CFD simulation for Ferry 2, both the bow thrusters operating with 130 kN thrust each during the period of time of 60 s.

**Figure 7 sensors-22-09133-f007:**
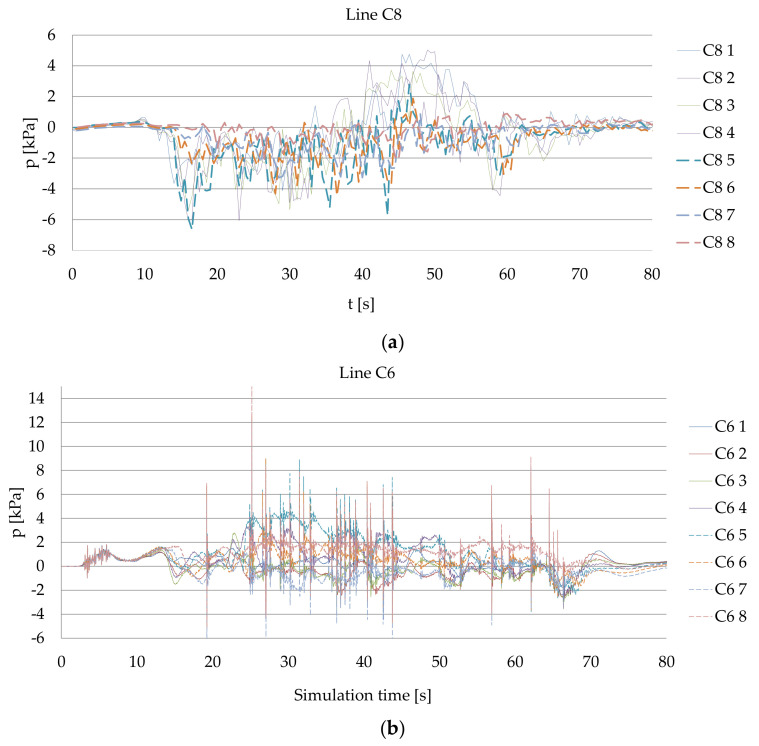
Pressure during the unberthing: (**a**) real measurements for Ferry 2 in the C8 line; (**b**) CFD simulation for Ferry 2 results in the C6 line corresponding to the C8 line during the measurements, both the bow thrusters operating with 130 kN thrust each during the period of time of 60 s.

**Figure 8 sensors-22-09133-f008:**
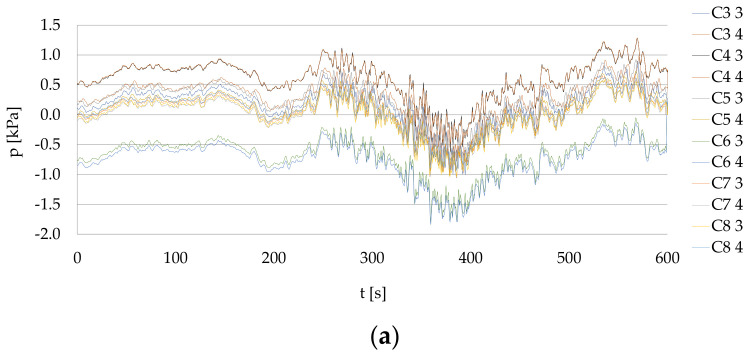
Pressure during the berthing: (**a**) real measurements collected during the berthing of Ferry 2; (**b**) maximum absolute values of pressure: P3, P4—measurements during the berthing of Ferry 2 in the C3–C8 lines using the pressure meters P3 and P4; P3 CFD and P4 CFD—results of the CFD simulation with both the bow thrusters operating with 105 kN thrust each during the period of time of 5 s; P3 CFD max and P4 CFD max–maximum single pressure pulses.

**Figure 9 sensors-22-09133-f009:**
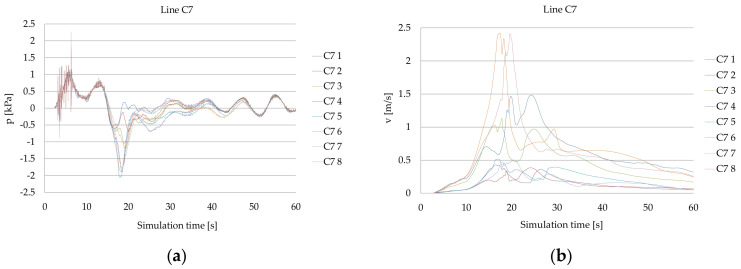
Results of the CFD simulation for Ferry2—both the bow thrusters operating with 105 kN thrust each during the period of time of 10 s: (**a**) pressure in the C7 line; (**b**) velocity on the quay wall in the C7 line.

**Figure 10 sensors-22-09133-f010:**
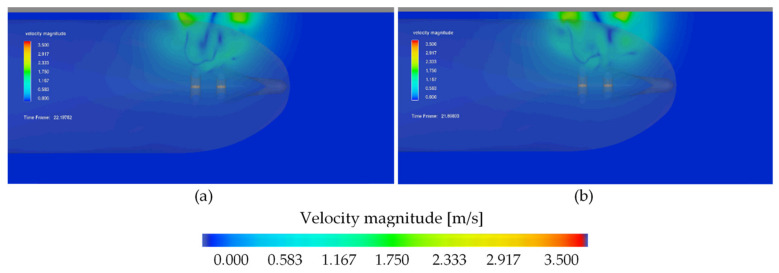
Velocity field on the seabed—results of CFD simulation for Ferry 2: (**a**) both the bow thrusters operating with 105 kN thrust each during the period of time of 10 s, maximum velocity 2.4 m/s; (**b**) both the bow thrusters operating with 130 kN thrust each during the period of time of 10 s, maximum velocity 3.0 m/s.

**Figure 11 sensors-22-09133-f011:**
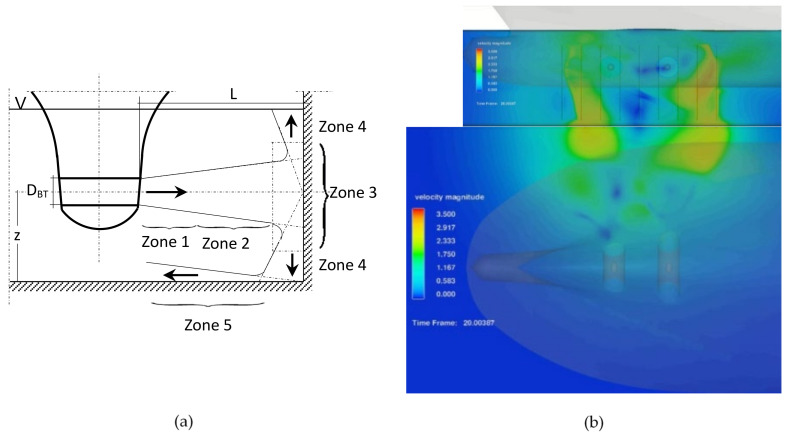
Bow thruster-generated flow distribution: (**a**) 5 zones of flow proposed by Schmidt cited in [8]; (**b**–**d**) CFD simulation—image of the highest velocity on the quay wall and on the seabed for both the bow thrusters operating with the thrusts equal to 105 kN, 130 kN and 160 kN during the period of time of 10 s, respectively.

**Figure 12 sensors-22-09133-f012:**
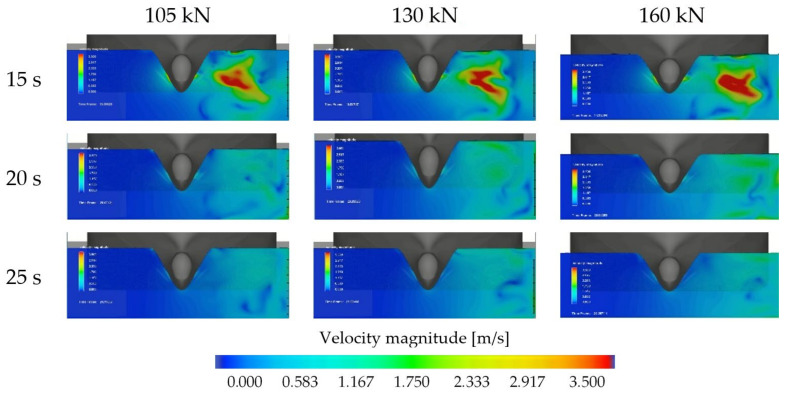
Bow thruster-generated flow distribution in the section of measuring C7 line during the time of simulation 15 s, 20 s and 25 s—both the bow thrusters operating for 10 s with thrusts of, respectively, 105 kN, 130 kN and 160 kN each.

**Figure 13 sensors-22-09133-f013:**
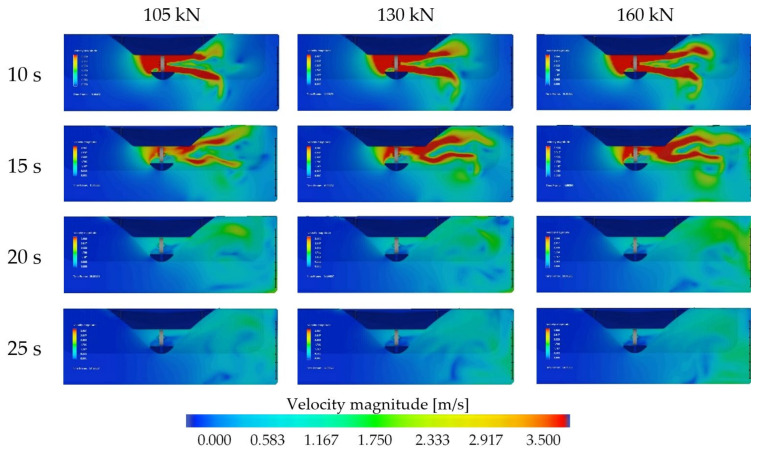
Bow thruster-generated flow distribution in the bow thruster symmetry plane between the C3 and C4 lines in 15 s, 20 s and 25 s of time simulation—both the bow thrusters operating with the thrusts 105 kN, 130 kN and 160 kN each, respectively, during the period of time of 10 s.

**Figure 14 sensors-22-09133-f014:**
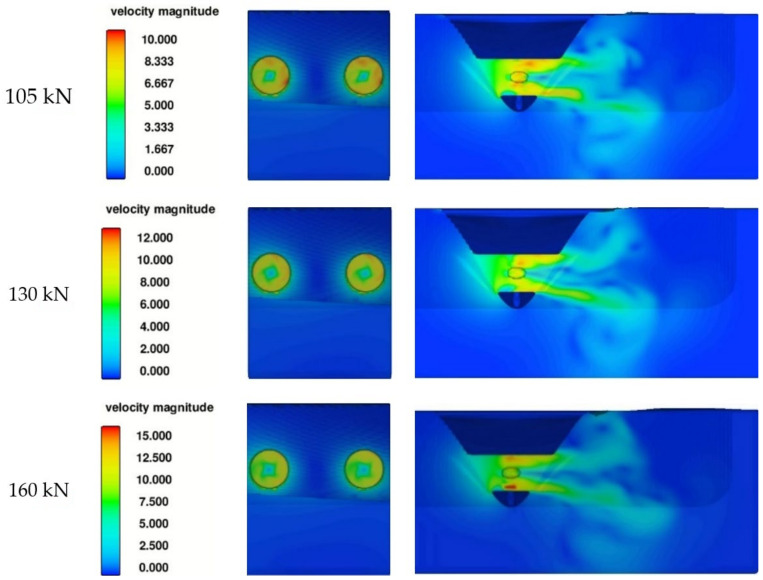
Efflux velocities for the thrust 105 kN, 130 kN and 160 kN, respectively.

**Table 1 sensors-22-09133-t001:** List of variables used in the paper.

Parameter	Description
B [m]	ship breadth
D_BT_ [m]	diameter of bow thruster
h [m]	water depth
L [m]	distance between the quay wall and bow thruster opening
L_oA_ [m]	ship length overall
N [kW]	main engines power
N_BT1_ [kW]	power of the first bow thruster from the bow
N_BT2_ [kW]	power of the second bow thruster from the bow
N_ST_ [kW]	power of stern thruster
n [rpm]	bow thruster propeller revolutions
p [kPa]	pressure
T [m]	ship draft
Thrust [kN]	thrust of the bow thruster
t [s]	time of measurements
t_s_ [s]	time of simulation
v [m/s]	flow velocity
v_0_ [m/s]	axial velocity of the bow thruster jet
z [m]	height of the bow thruster axis above the seabed
P [kg/m^3^]	water density

**Table 2 sensors-22-09133-t002:** Main particulars of the ferries.

Parameter	Ferry 1	Ferry 2
L_OA_ [m]	164.41	175.48
B [m]	27.69	30.3
T [m]	6.313	6.8
N [kW]	4 × 4840	4 × 7454
N_BT1_ [kW]	1275	1119
N_BT2_ [kW]	735	1119
N_ST_ [kW]	735	-

**Table 3 sensors-22-09133-t003:** Parameters of the virtual bow thruster–impeller properties.

Impeller Properities	
Accomodation coefficient for rotational velocity	14
Axial velocity coefficient	−1.5
Number of blades	4
Blade tip thickness in azimuthal direction [m]	0.05
Propeller revolution [rpm]/thrust [kN]	86/105; 94/130; 104/160
Bow thruster diameter [m]	2.37
Hub diameter [m]	0.68

**Table 4 sensors-22-09133-t004:** Maximum velocities generated by the bow thrusters on the seabed calculated for the axial velocity obtained from Equation (2) compared to the results of simulation and measurements.

Ferry	v_0_	v_max_			
	m/s	m/s			
	Equation (3)	German Method	Dutch Method	CFD Simulation	Measurement
Ferry 1	7.1	2.9	2.7	2.8	2.8
Ferry 2	6.7	2.5	2.6	2.4	2.4

**Table 5 sensors-22-09133-t005:** Maximum velocities generated by the bow thrusters on the seabed—CFD simulation results compared to the German and Dutch methods.

Thrust	v_0_	v_max_		
	m/s	m/s		
kN	CFD	CFD	German Method	Dutch Method
105	5.3	2.1	2.0	2.0
130	5.7	2.2	2.1	2.2
160	6.7	2.6	2.5	2.6

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
