# Peer review of "Ship Berthing and Unberthing Monitoring System in the Ferry Terminal"

_sensors, 2022, doi:10.3390/s22239133_

Round 1
Reviewer 1 Report
The manuscript deals with a study aimed at improving the safety of the mooring and useful for reducing maintenance costs.
In addition, the CFD method, based on simulation, allows you to check what happens when conditions change. So it proves useful for design purposes.
Therefore the manuscript, in addition to being written in a very clear and understandable way, proves useful in practice.
I only point out that figures 4, 10, 11, 12, 13 have a barely visible scale of values.
If the figures have the same scale of values ​​between them, then it is better to insert only one but visible one and remove all the others.
I understand that this is an output from the processing software, but inserting value scales where the values ​​are not very visible is of little use.
In these cases it would be better to prepare writings to overlap so that it improves visibility.
For everything else, I can only congratulate the authors and thank them for their work.
Author Response
Dear Reviewer,
Thank you for your comment., We really appreciate your help in improving the paper.
The visible scales are added in Figures 4, 10, 11, 12, 13.
Reviewer 2 Report
Review of the manuscript entitled ‘Ship berthing and unberthing monitoring system in the ferry terminal’
The monitoring system presented in the paper allows for the detection of exceedance of allow able loads. In the presented study the real ship hull form is modelled, including bow thrusters models in form of virtual propellers inside cylinders, with inlet and outlet openings diameters equal to the propeller ducts. The main objective of the study is to determine the bow thruster generated velocity field on the seabed on the basis of pressure measurement on the quay wall in time domain. The generally accepted empirical methods used in design practice of berth structures, seabed protections and development of operational guidelines were used to verify the numerical simulation results.
The study is interesting. However, I have some comments in the aim to improve the reading and understanding of the study.
In table 2 the parameters are not described
piezo-resistive pressure transmitters 131S give reference and supplier.
‘In this case Line C8 on the quay wall is corresponding to line C6 in simulation’ The sentence is not clear
No indication is given about the pressure P3 and P4 measurements
‘The relationship between dynamic pressure on the quay wall and velocity on the seabed can be calculated using Bernoulli equation.’
Give the relationship.
The figures are insufficiently commented for example the figure 12 or 13.
Author Response
Dear Reviewer,
Thank you for your comments, which help us to improve our paper. We can agree with all of them. We really appreciate your help in improving the paper.
Please find the corrections:
- In table 2 the parameters are not described
The missing description of N is added in Table 1:
“N [kW] - main engine power”
- piezo-resistive pressure transmitters give reference and supplier.
The reference and supplier is added in the text:
“(BCM Sensor Technologies www.bcmsensor.com)”
- ‘In this case Line C8 on the quay wall is corresponding to line C6 in simulation’ The sentence is not clear
This sentence has been removed from the text. The following explanation is added at the beginning of section 3.1, line 146:
“Bow thruster openings projections of Ferry 1 in moored position were between measuring lines C3, C4 and C6, C7. The bow thrusters projections of Ferry 2 were between lines C5, C6 and just after line C8 (Figure 1). In CFD simulation the positions of bow thrusters openings projections were between measuring lines C3, C4 and C6, C7. In case of Ferry 2 lines C3, C4 on the quay wall correspond to lines C5, C6 in CFD simulation and line C6 on the quay wall corresponds to line C8 in CFD simulation.”
- No indication is given about the pressure P3 and P4 measurements
The indication about the pressure P3 and P4 measurements is given. P3 and P4 measurements are included in Figure 8 (a) with added figure caption:
“Pressure during berthing: (a) real measurements collected during berthing of Ferry 2;”
The sentence in line 182 is changed to:
“The results of CFD simulation for Ferry 2 with both the bow thrusters operating with 105 kN thrust each during the period of time of 5 s are in line with the measurements collected during the berthing of Ferry 2 presented in Figure 8 (a). Figure 8 (b) presents the maximum absolute values of pressure within the C3 – C8 lines.”
- The relationship between dynamic pressure on the quay wall and velocity on the seabed can be calculated using Bernoulli equation. Give the relationship.
The sentence in line 230 is changed to:
The relationship between the dynamic pressure on the quay wall and velocity on the seabed (Equation 1) can be calculated based on Bernoulli equation (Equation 2), neglecting water viscosity.
Bernoulli equation is given in line 233
- The figures are insufficiently commented for example the figure 12 or 13.
The following comments are added:
line 238
“The flow velocity distribution presented in Figure 11 shows the maximum velocities generated on the quay wall and seabed after 20 s of simulation. The flow velocity distribution in time in dependence on the thrust applied to the bow thruster is presented in Figures 12 and 13.
The bow thruster generated flow distribution in the section of measuring C7 line during the period of time respectively 15 s, 20 s and 25 s for both the bow thrusters operating with the thrusts of 105 kN, 130 kN and 160 kN each is presented in Figure 12.”
Line 250:
“To determine the flow distribution with the maximum velocity values on the quay wall and seabed the simulation time should be long enough. In the studied case this time is about 20 s.
The bow thruster generated flow distribution in the bow thruster symmetry plane between C3 and C4 lines during the period of time respectively 10 s, 15 s, 20 s and 25 s for both the bow thrusters operating with the thrusts of 105 kN, 130 kN and 160 kN each presented in Figure 13 shows that during the first 10 s of simulation the maximum velocities are generated on the seabed only, then the jets reach the wall.”
Round 2
Reviewer 2 Report
The manuscript has been corrected and modified according to the comments of the reviewer and can now be published.